# PD-L1 Expression and Tumor-Infiltrating Lymphocytes in Thymic Epithelial Neoplasms

**DOI:** 10.3390/jcm8111833

**Published:** 2019-11-01

**Authors:** Rumi Higuchi, Taichiro Goto, Yosuke Hirotsu, Takahiro Nakagomi, Yujiro Yokoyama, Sotaro Otake, Kenji Amemiya, Toshio Oyama, Masao Omata

**Affiliations:** 1Lung Cancer and Respiratory Disease Center, Yamanashi Central Hospital, Yamanashi 400-8506, Japan; r-higuchi1504@ych.pref.yamanashi.jp (R.H.); nakagomi.takahiro@gmail.com (T.N.); dooogooodooo@me.com (Y.Y.); sotaro.otake@gmail.com (S.O.); 2Genome Analysis Center, Yamanashi Central Hospital, Yamanashi 400-8506, Japan; hirotsu-bdyu@ych.pref.yamanashi.jp (Y.H.); m-omata0901@ych.pref.yamanashi.jp (M.O.); 3Department of Pathology, Yamanashi Central Hospital, Yamanashi 400-8506, Japan; amemiya-bdcd@ych.pref.yamanashi.jp (K.A.); t-oyama@ych.pref.yamanashi.jp (T.O.)

**Keywords:** thymoma, thymic carcinoma, programmed death ligand 1 (PD-L1), CD8, immunotherapy

## Abstract

Thymic epithelial tumors (TETs) are rare malignant mediastinal tumors that are difficult to diagnose and treat. The programmed death 1 (PD-1) receptor and its ligand (PD-L1) are expressed in various malignant tumors and have emerged as potential immunotherapeutic targets. However, the immunobiology of TETs is poorly understood. We evaluated PD-L1 expression and the presence of tumor-infiltrating lymphocytes (CD8 and CD3 expression) in surgical TET specimens from 39 patients via immunohistochemistry and determined their relation to clinicopathological parameters. Cases with membranous reactivity of the PD-L1 antibody in ≥1% of tumor cells were considered positive. Positive PD-L1 expression was observed in 53.9% of cases. Histologically, PD-L1 expression was positive in 2/6 type A, 2/6 type AB, 3/9 type B1, 4/4 type B2, 5/6 type B3, and 5/8 type C TET cases. Thus, the number of cases with PD-L1 expression and the percent expression of PD-L1 were significantly higher in more aggressive thymomas (type B2 or B3). CD3+ and CD8+ tumor-infiltrating lymphocytes were diffusely and abundantly distributed in all cases. These data suggest that a PD-1/PD-L1 blockade is a promising treatment for TETs, with more beneficial treatment effects for aggressive thymomas such as type B2 or B3.

## 1. Introduction

Thymic epithelial tumors (TETs) are relatively rare malignant mediastinal tumors with an incidence of <1% of all adult cancers [1,2]. TETs are histologically classified as thymomas and thymic cancers. The World Health Organization (WHO) categorizes TETs into types A, AB, B1, B2, B3, and C (thymic cancer) according to the degree of aggressiveness (based on a histology) [3,4,5,6]. Complete resection is currently the only curative treatment [7,8,9], and the prognosis is poor for both unresectable and metastatic diseases [10,11]. Considering the limited activity of cytotoxic agents for advanced and refractory cases, the development of novel and innovative therapeutic strategies is essential, which requires a better understanding of thymic immunobiology.

Programmed death 1 (PD-1) is a coinhibitory inducible receptor present in T-cells and macrophages [12]. Tumor cells with an increased expression of the PD-1 ligand, PD-L1, are believed to escape host immunity through activation of the PD-1/PD-L1 pathway and the consequent suppression of effector immune responses [13,14,15,16]. The PD-L1 pathway is a unique coinhibitory pathway in cancer immunology, as it is selectively expressed in tumor cells but is very rarely in normal tissue, thereby making it a principal target for immunotherapy [16]. Novel agents resulting in an immune blockade have been introduced and are now widely used in the treatment of various cancers [17,18,19,20]. In particular, monoclonal antibodies targeting PD-1 or PD-L1 exert dramatic responses in many solid tumors, particularly melanoma, lung, and urothelial carcinomas [21,22,23,24]. Clinical trials evaluating the use of these agents in the treatment of thymic carcinoma (type C thymoma) are already underway and have demonstrated promising preliminary clinical outcomes [25,26]; however, in general, the validity, efficacy, and safety of treatment with immune checkpoint blockades have not been sufficiently investigated in TETs, including in other types of thymomas. With the emerging paradigm of immunotherapy in solid tumors, PD-L1 expression in TETs should be carefully studied and analyzed.

Accordingly, we evaluated the expression of PD-L1 and the presence of tumor-infiltrating lymphocytes (TILs) in thymic epithelial neoplasms (thymoma and thymic carcinoma) via immunohistochemical staining and analyzed the statistical associations between expression and clinicopathologic features. IL-10 mRNA was also measured to better characterize the aggressiveness of TET. Moreover, we discuss the possible clinical applications of immune checkpoint inhibitors for thymomas through a review of the published literature.

## 2. Material and Methods

### 2.1. Patients

A retrospective review was performed of tumor specimens available from 39 consecutive patients diagnosed with thymoma or thymic carcinoma undergoing surgical resection (*n* = 33) or biopsy (*n* = 6) at our hospital between January 2000 and October 2017. The clinicopathologic characteristics assessed included age, sex, histology, stage, smoking status, and diagnosis of myasthenia gravis (MG). The 2015 WHO classification was used for the histological classification of TET [27]. The study was of a retrospective design and was approved by the institute’s ethics committee.

### 2.2. Immunohistochemistry for PD-L1 and TILs

Specimens from 20 patients obtained between January 2000 and December 2013 were fixed with 20% nonbuffered formalin, and those obtained from 19 patients between January 2014 and October 2017 were fixed with 10% buffered formalin (Appendix A). Formalin-fixed paraffin-embedded tissues were sectioned at 5 μm, deparaffinized, rehydrated, and stained in an automated system (Ventana Benchmark ULTRA System; Roche, Tucson, AZ, USA) using commercially available detection kits and antibodies against PD-L1 (28–8, ab205921; Abcam, Tokyo, Japan), CD4 (4B12; Nichirei, Tokyo, Japan), CD8 (D1M8I; Cell Signaling Technology, Danvers, MA, USA), and CD3 (LN10; Leica Biosystems, Richmond, IL, USA). PD-L1 is primarily located in the cell membrane of tumor cells, and its expression was evaluated semi-quantitatively by two pathologists based on the proportion of PD-L1-positive tumor cells. A PD-L1 expression rate of 1% or greater was defined as PD-L1-positive. CD3 and CD8 were utilized as a marker for pan T lymphocytes and cytotoxic T lymphocytes, respectively. Thus, CD3+ and CD8+ lymphocytes were counted in the tumors, and the percentage of CD8+ lymphocytes in CD3+ lymphocytes was calculated as the TIL fraction, as previously reported [28,29,30].

### 2.3. mRNA Expression and Quantitative Real-Time PCR Analyses

RNA was isolated from frozen surgical specimens using an AllPrep DNA/RNA Mini Kit (Qiagen, Tokyo, Japan), and reverse transcription was performed using MultiScribe Reverse Transcriptase (Thermo Fisher Scientific, New York, NY, USA) in accordance with the manufacturer’s instructions. TaqMan™ Gene Expression Assays, including *IL10* (Hs00174086_m1), *PD-L1* (*CD274:* Hs00204257_m1), and *β-actin* (Hs99999903_m1), were purchased from Thermo Fisher Scientific. Quantitative real-time PCR analysis was performed using the ViiA™ 7 Real-Time PCR System (Thermo Fisher Scientific). The PCR reaction was performed with 10 μL TaqMan^®^ Fast Advanced Master Mix, 2 μL cDNA, 1 μL TaqMan™ Gene Expression Assay, and 7 μL nuclease-free water. Amplification reactions were performed in fast mode: 2 min at 50 °C and 20 s at 95 ℃ for denaturation, followed by 45 cycles of 1 s at 95 °C and 20 s at 60 °C. Gene expression was normalized to that of *β-actin*.

### 2.4. Statistical Analyses

Continuous variables are reported as means and standard deviations. Categorical variables were compared using the chi-squared test. Multivariate analyses were performed in JMP (SAS Institute, Cary, NC, USA). A two-tailed *p* < 0.05 denotes a statistically significant difference.

## 3. Results

### 3.1. Patient Characteristics

We studied surgical samples from 39 patients with thymic epithelial tumors who received surgery at our hospital between January 2000 and October 2017. Among the 39 patients, 21 were men and 18 were women, and 21 were smokers and 18 were nonsmokers. Histologically, there were six cases of type A, six cases of type AB, nine cases of type B1, four cases of type B2, six cases of type B3, and eight cases of thymic carcinoma (Appendix A). All thymic carcinomas were histologically classified as squamous cell carcinoma. The patients’ ages ranged between 23 and 85 (mean ± SD, 62.6 ± 14.6) years. One patient with type B thymoma exhibited comorbidity with MG (Case 22 in Table 1, Case 24 in Table 1).

### 3.2. Expression of PD-L1 in Tumor Cells

PD-L1 expression was positive (≥1%) in 2/6 patients with type A, 2/6 patients with type AB, 3/9 patients with type B1, 4/4 with type B2, 5/6 with type B3, and 5/8 patients with type C (Figure 1, Figure 2 and Figure 3, Table 1). In total, 51.6% of thymoma cases and 62.5% of thymic cancer cases stained positive for PD-L1. In type AB thymoma (*n* = 6), no cases stained positive for PD-L1 in the type A component of the tumor, and two cases stained positive for PD-L1 only in the type B component (Figure 1E,H), which indicates the possible presence of intratumor heterogeneity in PD-L1 staining.

A multivariate analysis (to identify factors that affect PD-L1 expression) revealed that only WHO classification was a significant determining factor of PD-L1 expression among age (*p* = 0.41), sex (*p* = 0.92), smoking habit (*p* = 0.67), WHO classification (*p* = 0.01), tumor size (*p* = 0.41), stage (*p* = 0.30), and formalin fixation method (*p* = 0.43). The number of cases with PD-L1 expression was significantly larger in thymoma types with more aggressive histology (type B2 or B3) in the WHO classification than in those with less aggressive histology (type A, AB, or B1) (Figure 4A). In addition, the percent expression of PD-L1 was significantly higher in type B2 or B3 thymoma than in thymomas with lower-grade histology (type A, AB, or B1) (Figure 4B).

### 3.3. mRNA Expression of PD-L1 and IL-10 in Tumor Tissues

PD-L1 mRNA expression was measured in tumor tissues by using real-time PCR. PD-L1 mRNA expression was higher in type B2, B3, and C TETs than in the other types (Figure 5A), which was consistent with the immunohistochemical findings. As an M2-type cytokine, IL-10 expression was also measured. IL-10 mRNA expression was the highest in type C TET and was significantly higher in higher grades of TET (types B2, B3, and C) than in the lower grades (types A, AB, and B1) (Figure 5B).

### 3.4. CD8 Expression in Lymphocytes

In all TET samples, CD3+ and CD8+ TILs were abundantly present in the tumors except for one case (Case 30 in Table 1), with 5% frequency of CD8+ expression in CD3+ lymphocytes. The percentage of CD8+ lymphocytes among the CD3+ lymphocytes in the tumors was high, at 82.8% ± 18.3% (mean ± SD) (Table 1).

### 3.5. CD4-Positive Lymphocytes in Tumor Tissues

CD4-positive lymphocytes were counted in 400× magnification fields under a microscope. The counts were higher in the type B1–3 thymomas than in types A, AB, or C (Figure 6).

## 4. Discussion

Immune checkpoint blockades are an emerging and hopeful treatment strategy for a variety of cancers [18,31,32]; however, only a subset of patients sustain clinical responses, and establishing a predictive marker for a good response remains challenging. In this context, we have begun to profile TETs for PD-L1 expression and additional immunotherapeutic markers. Here, we found positive PD-L1 expression (≥1%) in 53.9% of all surgical TET samples. None of the clinical parameters, including age, sex, smoking habit, tumor size, and stage, were correlated with PD-L1 immunoexpression, whereas there was a significant association between PD-L1-positive status and more aggressive thymomas (types B2 and B3). Furthermore, CD8+ TILs were abundantly detected in almost all TETs examined, supporting the use of an immune checkpoint inhibitor for TET treatment.

In our study, the expression of IL-10, as one of the M2-type cytokines, was measured by real-time PCR, which revealed that type B2–3 thymomas exhibited a higher expression of IL-10 than the other types of thymomas did. In type B2–3 thymomas, anti-inflammatory mediators are upregulated inside the tumor, thereby inducing a suitable microenvironment for the proliferation of tumor cells. In addition, the number of CD4+ lymphocytes in the tumor tissue was significantly higher in type B1–3 thymomas than in the other types. The abundance of both CD4+ lymphocytes and CD8+ lymphocytes in type B1–3 thymomas suggested that these lymphocytes surrounded the tumor cells and were ready to attack the tumor cells. These data also supported the potential of PD1/PD-L1 blockades as a promising treatment for aggressive thymomas, such as type B2 or B3.

According to the WHO classification, thymomas are classified into types A and B, while thymic carcinomas are categorized as type C. Although thymic carcinomas have cytological characteristics of malignant tumors, thymomas are essentially benign tumors cytologically. Thus, thymic carcinomas tend to be more infiltrating and rapidly aggressive than thymomas. Several studies have reported that PD-L1 expression was upregulated in types B2 and B3, but not in type C [33,34], which is consistent with our findings. This low PD-L1 expression in type C may partly reflect the fact that pathogenesis, histopathology, and biological phenotypes are totally different between types A and B and type C TETs. In addition, a smaller number of CD4+ lymphocytes in type C compared to type B can be attributed to the difference in lymphocyte richness in the microenvironment between types B and C.

Sufficient therapeutic effects of PD-1/PD-L1 inhibitors for TETs have been demonstrated in individual case reports, contributing to the promotion of clinical trials for advanced TETs [35,36]. Subsequently, some phase I clinical trials have used PD-1/PD-L1 inhibitors for various solid malignant tumors, including TETs, demonstrating acceptable clinical efficacy and safety despite the limited number of recruited patients [37,38]. Two phase II, single-center trials evaluating the clinical feasibility of pembrolizumab for TETs have also recently been reported [25,39]. Giaccone et al. [25] investigated 40 thymic carcinoma patients with disease progression after at least one line of chemotherapy: patients with autoimmune disease or other malignancies were excluded. They found an overall response rate of 22.5%: 1 patient achieved a complete response, 8 achieved a partial response, and 21 achieved stable disease. The toxicity was reported to be well tolerated, and fatigue, diarrhea, and fever were the major side effects noted. However, six (15%) of the patients developed serious autoimmune disorders, and two (5%) patients experienced severe polymyositis and myocarditis that required pacemaker implantation [25]. The other phase II trial, conducted by Cho et al. [39], targeted 33 TET patients (26 with thymic carcinomas and 7 with thymomas). The eligibility criteria included progression after at least one line of platinum-based chemotherapy, and patients with active autoimmune disease were excluded. The response rate in patients with thymic carcinoma and thymoma was 19.2% and 28.6%, respectively. However, various grade 3 or higher immunological side effects were observed in 5 (71.4%) of the 7 patients with thymoma and in 4 (15.4%) of the 26 patients with thymic carcinoma, including hepatitis, myocarditis, MG, thyroiditis, glomerulonephritis, colitis, and subcutaneous myoclonus [39].

Overall, our study suggests that the application of immunotherapy in TET treatment would be lucrative, since PD-L1 positivity was diffuse, strong, and present in a large number of TETs (53.9%), as confirmed in previous reports [33,34,40,41,42,43,44,45,46,47,48,49,50]. To the best of our knowledge, there have been 14 published studies, including the present study, on PD-L1 expression in TETs (Table 2) [33,34,40,41,42,43,44,45,46,47,48,49,50]. Although only a limited number of patients were included in each study due to the fact that TETs are a rare disease, PD-L1 expression was assessed in a total of 801 thymomas and 180 thymic carcinomas, with a wide range in positivity of 18%–92% and 36%–88%, respectively. One possible explanation for these variable findings between studies is the use of different antibodies for immunohistochemistry detection, which could have resulted in variations in the detected staining site, pattern, and intensity of PD-L1. In addition, the studies used different cut-off scores for positivity determinations of PD-L1 expression. Different FDA-approved assays are available for the determination of PD-L1 expression, including the 28–8 assay used in the present study, to select patients with lung cancer for immunotherapy. However, there is no consensus for the most appropriate assay for PD-L1 testing in TETs. Nevertheless, most of these studies also found a correlation of PD-L1 expression with type B histology [33,34,41,42,43,44,47,49,50] (in concordance with our results), which suggests a nonredundant application of immunotherapy for these aggressive types of thymomas.

Apart from PD-1/PD-L1 blockade therapy, recent studies have identified a missense mutation (chromosome 7 c.74146970T>A) in the *GTF2I* gene at a high frequency in TETs [51,52]. The recurrent presence of this point mutation in TETs may represent a target for potential therapeutic use, and novel treatment strategies, such as molecular-targeted therapy, gene therapy, or CAR T-cell therapy, are promising for the future.

Our study is certainly limited by its relatively small sample size; thus, a larger series will be needed to more comprehensively evaluate the immunophenotypic landscape of TETs and more clearly elucidate associations with clinical parameters in a more comprehensive multivariate analysis. However, as the major aim of this preliminary analysis was to examine therapeutic targets that should be prioritized for clinical development, the modestly sized cohort can still provide useful insight. The development of immune checkpoint inhibitors in the treatment of TETs represents an important next step in improving outcomes in this otherwise difficult-to-treat disease. Here, we demonstrated that the tumor cells were prevalently surrounded by abundant CD8+ TILs, suggesting that the inhibition of an immune evasion via PD-1/PD-L1 would allow for a strong killing effect specific to tumor cells [53,54,55,56]. This novel finding of the presence of abundant TILs should promote and expedite clinical trials to establish immunotherapies for TETs.

## 5. Conclusions

The high expression of PD-L1 and abundant CD8+ lymphocytes in TETs provides a theoretical foundation for the clinical adoption of an immunotherapy strategy for targeting PD-1/PD-L1. Clinical trials should be expedited with careful monitoring of adverse effects to best explore the potential immunotherapies and thus improve the outcome of treatment for patients with TETs.

## Figures and Tables

**Figure 1 jcm-08-01833-f001:**
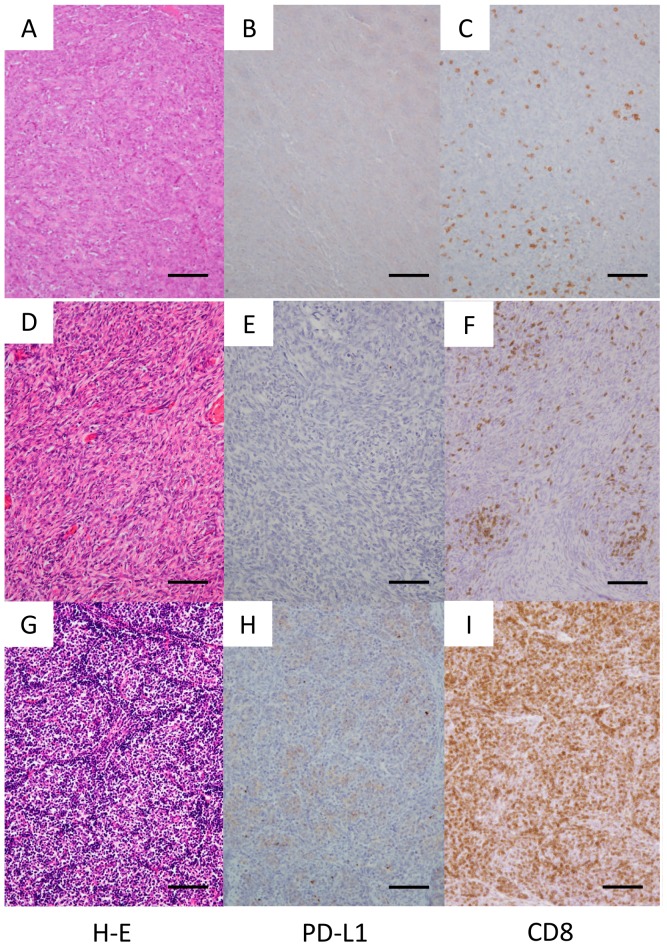
Type A and type AB thymoma. Representative cases are shown. (**A**–**C**) Type A thymoma: Case 2 in Table 1. (**D**–**I**) Type AB thymoma: Case 10 in Table 1. (**D**–**F**): Type A component of type AB thymoma. (**G**–**I**): Type B component of type AB thymoma. Original magnification, 100×; scale bar, 100 μm.

**Figure 2 jcm-08-01833-f002:**
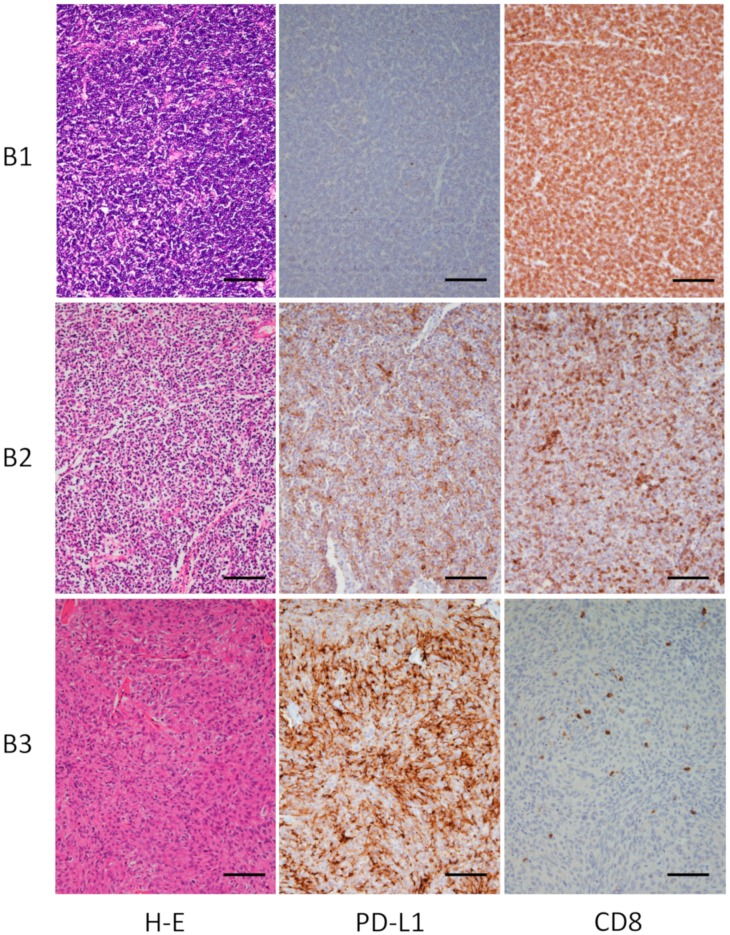
Histological and immunohistological findings for type B thymoma. Representative cases are shown. Type B1 thymoma: Case 21 in Table 1; Type B2 thymoma: Case 23 in Table 1; Type B3 thymoma: Case 31 in Table 1. Original magnification, 100×; scale bar, 100 μm.

**Figure 3 jcm-08-01833-f003:**
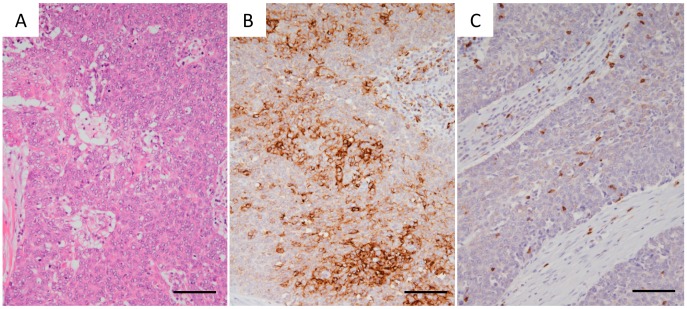
Histological and immunohistological findings for thymic carcinoma. A representative case (Case 38 in Table 1) is shown. (**A**) Hematoxylin and eosin staining. (**B**) PD-L1 Immunohistochemical staining. (**C**) CD8 immunohistochemical staining. Original magnification, 100×; scale bar, 100 μm.

**Figure 4 jcm-08-01833-f004:**
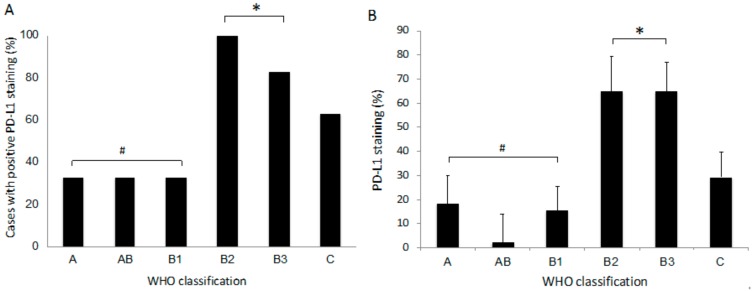
Comparison of PD-L1 immunohistochemical staining according to the World Health Organization (WHO) classification of TET. (**A**) Cases with positive PD-L1 staining. (**B**) PD-L1 staining score. * *p* < 0.05 compared to lower grades (#).

**Figure 5 jcm-08-01833-f005:**
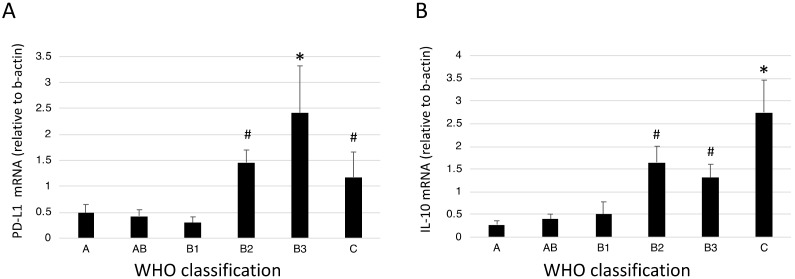
*PD-L1* and *IL-10* mRNA expression according to the WHO classification of TET. (**A**) *PD-L1* expression. (**B**) *IL-10* expression. RNA was isolated and mRNA expression was quantified with regard to mRNA expression of *β-actin* by using real-time PCR. Error bars represent the SDs of the means. * *p* < 0.05 compared to all other TETs. ^#^
*p* < 0.05 compared to lower grades (types A, AB, and B1).

**Figure 6 jcm-08-01833-f006:**
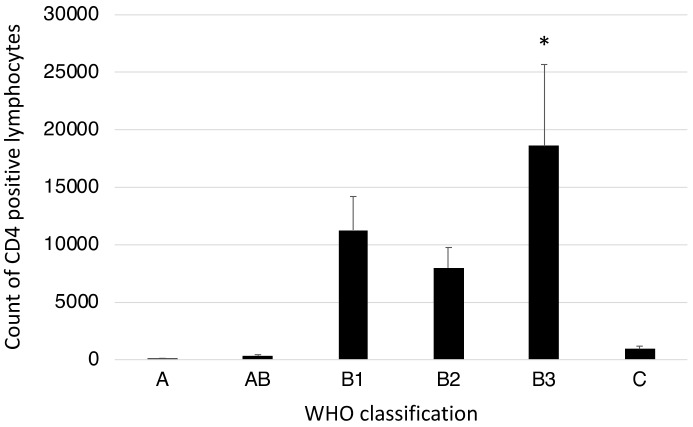
Comparison of CD4+ lymphocyte counts according to the WHO classification of TETs. * *p* < 0.05 compared to type A, AB, and C TETs.

**Table 1 jcm-08-01833-t001:** Immunohistochemical findings for PD-L1 and CD8/CD3 in each patient.

Pathology	Case No.	PD-L1 Expression	CD8/CD3
A (*n* = 6)	1	(-)	90%
	2	80%	90%
	3	(-)	90%
	4	(-)	90%
	5	(-)	90%
	6	30%	90%
AB (*n* = 6)	7	(-)	90%
	8	(-)	90%
	9	(-)	90%
	10	A(-)/B:3%	90%
	11	(-)	90%
	12	A(-)/B:10%	90%
B1 (*n* = 9)	13	(-)	70%
	14	70%	90%
	15	(-)	90%
	16	(-)	50%
	17	70%	90%
	18	(-)	90%
	19	1%	90%
	20	(-)	95%
	21	(-)	90%
B2 (*n* = 4)	22	70%	90%
	23	70%	90%
	24	50%	90%
	25	70%	90%
B3 (*n* = 6)	26	(-)	90%
	27	70%	70%
	28	80%	40%
	29	60%	90%
	30	90%	5%
	31	90%	90%
Ca (*n* = 8)	32	40%	40%
	33	(-)	90%
	34	90%	90%
	35	(-)	90%
	36	(-)	90%
	37	30%	90%
	38	70%	70%
	39	5%	90%

(-): means negative staining.

**Table 2 jcm-08-01833-t002:** Summary of previously published studies of PD-L1 immunohistochemistry in thymomas.

No.	Author/Year	No. of Thymoma/Thymic Ca Cases	Sex Ratio (M/F)	MG	Clone of PD-L1 (Dilution)	Criteria for Positivity: Cut-Off	Percent Positivity	Most Common Histologies with PD-L1 Positivity
1	Brown et al. [40]/2003	26/8	NA	NA	29E.2A3, 29E.5A9	Not defined	Thymoma: 81%, Ca: 88%	Not defined
2	Padda et al. [47]/2015	65/4	1.1:1	16	5H1 murine mAb (1:100) 15 rabbit mAb (1:1000)	Intensity of staining (score 0–3): score 3	Thymoma: 68%, Ca: 75%	B2, Ca
3	Katsuya et al. [45]/2015	102/37	0.6:1	NA	E1L3N (1:800)	H-score (staining intensity (0–3) × % of positive cells (0%–100%)): score 3	Thymoma: 23% Ca: 70%	Ca
4	Yokoyama et al. [50]/2016	82/0	0.6:1	18	EPR1161 (1:200)	Youden’s index: > 38%	Thymoma: 53.7% Ca: Not available	B2 and B3
5	Enkner et al. [42]/2017	37/35	NA	NA	E1L3N	H-score (cut-off not defined)	Thymoma: 76% in B3, 13% in A Ca: 53%	B3, Ca
6	Tiseo et al. [48]/2017	87/20	1.2:1	26	E1L3N (1:500)	H-score: Score 3	Thymoma: 18% Ca: 65%	Ca
7	Marchevsky et al. [34]/2017	38/8	NA	NA	SP142 (1:250)	Membranous expression ≥ 6% tumor cells	Thymoma: 92% Ca: 50%	B2, B3
8	Weissferdt et al. [49]/2017	74/26	1.3:1	19	EPR4877 (1:250)	Strong membranous staining: > 5% tumor cells	Thymoma: 64% Ca: 54%	B3
9	Arbour et al. [33]/2017	12/11	1.3:1	2	E1L3N	Membranous expression > 25% tumor cells	Thymoma: 92% Ca: 36%	B2, B3
10	Owen et al. [46]/2018	32/3	1.1:1	NA	22C3	Intensity of staining (score 0–5): Score 1	Thymoma: 81% Ca: 100%	Not associated
11	Chen et al. [41]/2018	50/20	0.8:1	1	SP142	Membranous expression ≥ 5% tumor cells	Thymoma: 48% Ca: 70%	B3, Ca
12	Guleria et al. [43]/2018	84/0	1.5:1	28	SP263	Membranous expression > 25% tumor cells	Thymoma: 82%	B1, B2, B3
13	Hakiri et al. [44]/2019	81/0	1:1	17	SP142 (1:50)	Membranous expression ≥ 1% tumor cells	Thymoma: 27%	B2, B3
14	Present study	31/8	1.2:1	1	28–8	Membranous expression ≥ 1% tumor cells	Thymoma: 51.6% Ca: 62.5%	B2, B3

Abbreviations: PD-L1, programmed death ligand 1; M, male; F, female; MG, myasthenia gravis; +ve, positive; Ca, carcinoma; Ab, antibody; mAb, monoclonal antibody; PD-1, programmed death 1; NA, not available.

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
