# Peer review of "PD-L1 Expression and Tumor-Infiltrating Lymphocytes in Thymic Epithelial Neoplasms"

_jcm, 2019, doi:10.3390/jcm8111833_

Round 1
Reviewer 1 Report
Interpretation of figure 1 and its legend is a bit confusing. I suggest to modify it according to figure 2 that is clearer.
In Figure 5, b-actin should be corrected in β-actin.
Author Response
Interpretation of figure 1 and its legend is a bit confusing. I suggest to modify it according to figure 2 that is clearer.
Response: We changed the format of Figure 2, according to the reviewer’s suggestion.
In Figure 5, b-actin should be corrected in β-actin.
Response: We corrected “b-actin” to “β-actin”, as the reviewer suggested.
Thank you very much for your thoughtful comments.
Reviewer 2 Report
This paper entitled “PD-L1 Expression and Tumor-Infiltrating 2 Lymphocytes in Thymic Epithelial Neoplasms” by Higuchi et al., described the PDL-1 expression in thymic epithelial tumor (TET). They evaluated the PD-L1 expression and the presence of tumor-infiltrating lymphocytes (CD8 and CD3 expression) in surgical TET specimens from 39 patients via immunohistochemistry, and determined their relation to clinicopathological parameters.
Major concerns:
The thymus provides an inductive environment for development of T cells from hematopoietic progenitor cells. The ratio of CD8/CD3 to be used to represent the proportion of TIL in thymus in this study is need to more concerned. Obviously, the data “CD3/CD8 positive in almost all patients” in this study has a big risk to discuss the correlation between PDL-1 expression level and T cells infiltration in TET tumor sections. They must be clearly explain why it can represent the proportion of TIL and provide related references to support. (Fig 5)Authors need to discuss why evaluate IL-10 expression in this study. There is no mention of IL-10 in introduction, the meaning must be clarified. (Fig 4) Authors mentioned that the more malignant thymoma, the stronger the performance of PD-L1, so the expression of PD-L1 of type B2,3 much higher than type A, AB, B1. But this does not explain why type C PD -L1 performance will be weaker. According to the theory in the article, the most malignant type C, its PD-L1 performance should be the strongest. It is better to explain the possible inferences of type C in the discussion. (Fig 6) The same question with figure 4, why the most malignant type C, its CD4+ lymphocyte performance is weakened, it is also best to propose some possible inferences.
Minor concerns:
(Fig 1-3) The magnification of all pathology figures is not mentioned; also, the scale of each figure is not indicated. (Line 153) “5% frequency “changed to “5% frequency of CD8+ expression in CD3+ lymphocytes” wold be more easy to be understand. In discussion, too many side effects discussion of immune therapy to be mentioned, I suggest to delete some and more focusing on the main results of this article. Owing to the PDL-1 expression result of type C is not good, I suggest authors should provide more other therapeutic strategies other than PD-1-PDL-1 axis blockade in TET treatment in discussion.
Author Response
Major concerns:
The thymus provides an inductive environment for development of T cells from hematopoietic progenitor cells. The ratio of CD8/CD3 to be used to represent the proportion of TIL in thymus in this study is need to more concerned. Obviously, the data “CD3/CD8 positive in almost all patients” in this study has a big risk to discuss the correlation between PDL-1 expression level and T cells infiltration in TET tumor sections. They must be clearly explain why it can represent the proportion of TIL and provide related references to support.
Response: In many scientific studies, CD8 and CD3 are utilized as a marker for cytotoxic T lymphocytes and pan T lymphocytes, respectively. Also, TIL ratio was defined as CD8+/CD3+ in previous reports (Cermakova P et al. Anticancer Res 2014:34: 5555-5562, Xiang Hu et al. EBioMedicine 2018:35:178–188; Yu A et al. PLoS One 2018:13(10):e0205746.)
We added these descriptions and references in the Material and Methods section.
(Fig 5) Authors need to discuss why evaluate IL-10 expression in this study. There is no mention of IL-10 in introduction, the meaning must be clarified.
Response: IL-10 was measured in each type of thymoma, on the hypothesis that M2 cytokine secretion by tumor-associated macrophages may be linked to a more aggressive phenotype in thymomas.
We added this description in the Introduction section.
(Fig 4) Authors mentioned that the more malignant thymoma, the stronger the performance of PD-L1, so the expression of PD-L1 of type B2,3 much higher than type A, AB, B1. But this does not explain why type C PD -L1 performance will be weaker. According to the theory in the article, the most malignant type C, its PD-L1 performance should be the strongest. It is better to explain the possible inferences of type C in the discussion.
Response: According to the WHO classification, thymomas are classified into type A and B, while thymic carcinomas are categorized as type C. Although thymic carcinomas have cytological characteristics of malignant tumors, thymomas are essentially benign tumors cytologically. Thus, thymic carcinomas tend to be more infiltrating and rapidly aggressive than thymomas.
Several studies reported that PD-L1 expression was upregulated in type B2 and B3, but not in type C (Arbour KC et al. PLoS One 2017:12:e0182665; Marchevsky AM et al. Hum Pathol 2017:60:16-23), which is consistent with our findings. I would like to guess why type C expressed less PD-L1 than type A and B, but no scientific explanation to that question is possible, except for the fact that pathogenesis, histopathology and biological phenotypes are totally different between type A-B and C TETs.
We added these descriptions in the Discussion section.
(Fig 6) The same question with figure 4, why the most malignant type C, its CD4+ lymphocyte performance is weakened, it is also best to propose some possible inferences.
Response: Histologically, type B thymoma is rich in lymphocytes, whereas type C isn’t. These basic differences in the tumor microenvironment may explain the smaller number of CD4+ lymphocytes in type C.
We added this description in the Discussion section.
Minor concerns:
(Fig 1-3) The magnification of all pathology figures is not mentioned; also, the scale of each figure is not indicated.
Response: Magnification of the microscopic observation was described in the figure legends, and the scale bars were put in the microscopic views.
(Line 153) “5% frequency “changed to “5% frequency of CD8+ expression in CD3+ lymphocytes” wold be more easy to be understand.
Response: The phrase was revised, according to the reviewer’s suggestion.
In discussion, too many side effects discussion of immune therapy to be mentioned, I suggest to delete some and more focusing on the main results of this article.
Response: We agree with the reviewer and deleted one paragraph (line 201-212), because these discussions might be slightly deviated from the main topic of our paper.
Owing to the PDL-1 expression result of type C is not good, I suggest authors should provide more other therapeutic strategies other than PD-1-PDL-1 axis blockade in TET treatment in discussion.
Response: In order to establish novel treatment strategies, the identification of targetable gene mutations is an emerging need. Recent studies identified a missense mutation (chromosome 7 c.74146970T>A) in GTF2I gene at high frequency in TETs (Petrini I et al. Nat Genet 2014:46:844-849; Radovich M et al. Cancer Cell 2018:33:244-258). The recurrent presence of this point mutation in TETs may represent targets of potential therapeutic use, and novel treatment strategies, such as molecular-targeted therapy, gene therapy and CAR T-cell therapy, are promising in the future.
We added these descriptions in the Discussion section.
Our paper has been much improved by the reviewer’s suggestions. Thank you very much for your thoughtful comments.
This manuscript is a resubmission of an earlier submission. The following is a list of the peer review reports and author responses from that submission.
Round 1
Reviewer 1 Report
The study is a preliminary analysis of PD-1 markers in thymomas and CD8+ markers. Based on the previous literature and their own observation by immunostaining for PD-1 and CD8 markers, the authors have indicated immunotherapy as a promising therapy for most thymoma with high PD-1 expression.
If, blocking the PDL-1 in any of these cases and then assay for lower tumorogenic properties could be shown then that would strengthen the manuscript.
Reviewer 2 Report
In this study Higuchi et. al reports the correlation between PDL1 expression in thymic epithelial neoplasms and CD3+/CD8+ TIL infiltration. The paper is interesting and has data from valuable patient samples but needs further extensive studies before it can be considered fit for publication:
1) The authors should look into CD4+ cells in the sections as well. It is known that CD4+ cells also express PD-1. Thus should play a role in the observed PDL-1.
2) Along with PDL1 expression the authors should look into PD-1 expression intensity within the CD3/CD8+ and CD3/CD4+ cells in the samples.
3) Along with showing the protein level the authors should consider performing qPCR using these tissues from different tumors grades to show the difference in PDL-1 expression.
4) Did the authors find any changes in Treg cells (Foxp3+) between the different tumor grades?
5) The authors should look for differences in M2-type cytokine levels (CD10, CD4, CD13 etc) using western blots/ELISA etc. to justify their finding.
6) Did the authors find correlation between tumor grades, PDL-1 expression of T cells and 'M2-ness' of the tumor infiltrating macrophages? Exploring this can produce a probable mechanism of worse tumor grade which can probably be linked to greater tumor induced-immunosuppression.